# Measurement and 3D Visualization of the Human Internal Heat Field by Means of Microwave Radiometry

**DOI:** 10.3390/s21124005

**Published:** 2021-06-10

**Authors:** Igor Alexandrovich Sidorov, Alexsandr Grigorevich Gudkov, Vitalij Yurievich Leushin, Eugenia Nikolaevna Gorlacheva, Eugenij Pavlovich Novichikhin, Svetlana Victorovna Agasieva

**Affiliations:** 1RL Research Institute, Bauman Moscow State Technical University, 105005 Moscow, Russia; profgudkov@gmail.com (A.G.G.); ra3bu@yandex.ru (V.Y.L.); gorlacheva@yandex.ru (E.N.G.); 2Fryazinsky Branch of the V. A. Kotelnikov IRE of the Russian Academy of Sciences, 1141190 Moscow, Russia; epnov@mail.ru; 3Academy of Engineering, RUDN University, 117198 Moscow, Russia; s.agasieva@mail.ru

**Keywords:** microwave radiothermograph, non-invasive temperature measurement, malignant neoplasm, 3D visualization, antenna-applicator, interpolation

## Abstract

The possibility of non-invasive determination of the depth of the location and temperature of a cancer tumor in the human body by multi-frequency three-dimensional (3D) radiothermography is considered. The models describing the receiving of the human body’s own radiothermal field processes are presented. The analysis of the possibility of calculating the desired parameters based on the results of measuring antenna temperatures simultaneously in two different frequency ranges is performed. Methods of displaying on the monitor screen the three-dimensional temperature distribution of the subcutaneous layer of the human body, obtained as a result of data processing of a multi-frequency multichannel radiothermograph, are considered. The possibility of more accurate localization of hyperthermia focus caused by the presence of malignant tumors in the depth of the human body with multi-frequency volumetric radiothermography is shown. The results of the study of various methods of data interpolation for displaying the continuous intrinsic radiothermal field of the human body are presented. Examples of displaying the volumetric temperature distribution by the moving plane method based on digital models and the results of an experimental study of the thermal field of the human body and head are given.

## 1. Introduction

As a rule, thermal anomalies appear simultaneously with pathological processes occurs inside the human body distorting the natural heat field inside the body and on its surface, which can be detected using the method of microwave radiometry [1,2,3,4,5,6,7,8,9,10,11,12,13,14,15,16,17]. In this way, breast cancer [18,19], various pathologies of the brain [12,13,20,21,22], coronary heart disease [23], arthritis and blood flow disorders [24,25], urological diseases [26] and some others [27,28,29,30] can be detected. For further microwave radiometry, methods developing various microwave radiometers have been developed [9,21,31,32,33,34], using both their elements [35,36,37] and application methods with various models [38,39,40].

The various external influences can distort the natural heat field of a body, for example, drug exposure, physiological effects, the effect of a laser, infrared or microwave radiation. Registration of dynamic changes in the body’s own heat field allows reliable diagnosis of various diseases [1,2,18,20,38,39]. The temperature of the body surface can be accurately measured by medical thermometers, infrared pyrometers, thermal cameras and other devices. However, these devices cannot measure the internal temperature of the body. The introduction of a thermal sensor under the skin distorts the natural heat field and can cause damage to internal organs.

Therefore, there is an urgent need to improve non-invasive methods of measuring and visualizing the 3D distribution of internal temperatures in the depth of the human body with the aim of early diagnostics and monitoring of various pathological processes.

A special antenna-applicator is installed on the surface of the human body to measure the temperature with a radiometer [35,36], which is harmonized with the body at the installation point. In this case, the temperature averaged by the antenna pattern formed by the antenna-applicator inside the human body is measured with an accuracy of better than one tenth of a degree Celsius. To monitor the thermal processes dynamics inside the human body, it is necessary to measure the internal temperatures of the body simultaneously at several points of the body for a certain period of time, usually from 10 minutes to one hour. This problem is solved by using a multi-channel radiothermograph, which has several antenna-applicators, the signals from which are fed to the input of the radiometer through a specialized microwave switch. Each antenna-applicator receives its own electromagnetic field radiated by the body within the antenna pattern and from different depths. The radiation generated in the depth of the human body, spreading to the surface, is partially attenuated due to absorption in human tissues. The amount of wave attenuation depends on the type of tissue (muscle tissue, adipose tissue, bone tissue, skull, brain) and on the wavelength. Numerically, the attenuation is characterized by the value of the skin layer or the depth at which the power of the electromagnetic wave decreases by a factor of e (2.7282). The size of the skin layer depends on the wavelength of the radiation. Therefore, for the F2 range with a wavelength of 43 cm, the value of the skin layer for breast tissue is about 7 cm, and for the F1 range with a wavelength of 21 cm, about 3.5 cm. Thus, it is possible to differentiate the location of the source of increased thermal radiation by depth by measuring the power of the body’s own thermal radiation at one point, but in different frequency bands. This is the methodological basis of 3D radiothermography.

The use of radiometers operating at different frequencies in the same system permits us to obtain information about the internal temperatures of body areas located at different depths [21], and potentially allows us to restore the 3D distribution of temperatures inside the body. The problem of constructing 3D images of the internal temperatures of the human body based on measuring the intensity of the body’s own thermal electromagnetic radiation is solved using a multi-channel multi-frequency radiothermograph.

It is shown in [41] that in order to implement an optimal combination of multi-frequency and multi-channel characteristics in one medical radiothermograph design, it is necessary to develop optimal antenna-applicator designs that provide the specified technical characteristics. Paper [42] presents the results of designing a printed broadband antenna for a multichannel multi-frequency radiothermograph. This antenna, with a diameter of 30 mm, in the form of a ring-shaped emitter made by printing, with a built-in infrared (IR) temperature sensor, is highly technologically advanced, has a wide operating frequency band and provides the possibility of simultaneous IR thermography. The article [43] presents the results of theoretical research related to the development of a small-sized ultra-wideband printed ring antenna with a diameter of 22 mm, designed to measure the microwave radiation of biological tissues in the frequency range of 1.4–4.6 GHz.

The purpose of this article is to show the possibility of 3D visualization of the internal thermal field of a person using the results of measurements of radio brightness temperatures, simultaneously measured at different points of the body, using a multi-channel multi-frequency radiothermograph. The novelty of the approach considered is an attempt to more accurately localize the heat source position at the antenna-applicator radiation pattern the main beam zone, due to 3D visualization of the internal heat field of a body and monitoring the temperature dynamics of a heat point source inside the human body.

The most important aspects of the hardware, interpolation and 3D visualization of the internal thermal human field algorithms, as well as the tests results of the new multichannel multi-frequency radiothermograph, are presented below.

## 2. Materials and Methods

A multichannel radiothermograph can be used in practice to detect malignant neoplasms in the early stages of the development of pathology, even when they have not yet been detected by the X-ray method. The method is based on a glucose test, when the patient is given to drink 30 g of an aqueous glucose solution on an empty stomach. Glucose, as a high-calorie substance, is absorbed and carried by the bloodstream throughout the body, feeding the cells [31]. At the same time, the body temperature briefly increases evenly by one to two tenths of a degree. If there is a malignant formation somewhere, then in the place of its localization the temperature rises sharply and the temperature contrast can be one or two degrees, which occurs for a short time, for about a few minutes. Analysis of the dynamic process of temperature changes in the places where the antenna-applicators are installed allows to diagnose and localize the position of the tumor. And for clarification the depth of the tumor location, it can be carried out based on the analysis of the data obtained by the method described.

The analysis can be carried out on the basis of a simplified model of thermal fields in the depth of the human body described in [3] where it is shown that the maximum detection depth of the heat source *z*_max_ depends on the value of the skin layer *z_s_* for a given frequency range, the thermal contrast Δ*T* of the source and the sensitivity of the radiometer *δT* by Formula (1):(1)$$z_{\max}=z_{s}\cdot \ln\left(\frac{\Delta T}{\delta T}\right)$$

It is also shown that if the source is detected in both frequency ranges, that is, the depth of the tumor is less than the corresponding values of *z*_max_, then the depth of the source *Z_c_* and its temperature *T_c_* can be calculated using Formula (2):(2)zc=zsλ1zsλ2zsλ1−zsλ2ln(Tbλ1−T0Tbλ2−T0)Tc=T0+Tbλ2−Tbλ1(Tbλ1−T0Tbλ2−T0)ZSλ1(ZSλ1−ZSλ2)−(Tbλ1−T0Tbλ2−T0)ZSλ2ZSλ1−ZSλ2
where *T*_0_ is the temperature of the body, *T_bλ_*_1_ and *T_bλ_*_2_-temperature radiothermography at the location of the tumor, *Z_Sλ_*_1_ and *Z_Sλ_*_2_—the size of the skin layer on the frequency bands F1 and F2. A graph of the calculated values of brightness temperatures at the location of the tumor, depending on the depth of the tumor in centimeters for the size of the skin layer of 3.5 cm when the thermal contrast of the tumor is 2 degrees, is shown in Figure 1.

The block diagram of a multi-channel dual-frequency radiothermography is shown in Figure 2. The radiothermograph consists of two identical five-channel UHF (Ultra High Frequency) and L-band radiometric receivers with applicator antennas, two identical signal pre-processing processors (PPP), and an information acquisition and transmission module (IATM) that transmits the collected information data to a personal computer for final processing and visualization of 3D images of the internal thermal field of a human body. The block diagram of a five-channel radiometric receiver is shown in Figure 3. The receiver consists of an input modulator built on a microwave switch chip, a microwave circulator, a multi-stage LNA, a band pass filter, a square power detector, a low-pass filter, and a low-pass amplifier. For the organization of scaterometric reception, the receiver has a noise generator on a noise diode and a noise signal switch.

The receiver operates under the control of the PPP module, which controls the modulation process. Control signals are sent to the control inputs of the switches. Switching of the receiving channels is carried out by transferring the control codes from the PPP module to the control inputs of the channel switch.

The modulation process is periodic with a period of approximately 10 milliseconds. The entire modulation period is divided into six equal parts, sub-periods. The first sub-period is designed to receive calibration signals from the matched load and the noise generator. Eighty percent of the sub-period is spent receiving by the signal from the matched load and 20% receiving the signal from the noise generator.

To set a one-to-one correspondence of the radiometer output signal to the internal temperature of a biological object, expressed in degrees on the Kelvin scale, the calibration procedure is used. Radiometers are characterized by two types of calibrations: internal, using reference sources of thermal noise built into the radiometric receiver, and external, using external equipment.

Internal calibration is performed by receiving calibration signals from internal noise sources. In the first sub-period of modulation, signals are received from the receiving channels of the antenna applicators, in the second sub-period from the first channel, in the third sub-period from the second channel, in the fourth sub-period from the third channel, in the fifth sub-period from the fourth channel, and in the sixth sub-period from the fifth channel. In the second sub-period, a signal from the antenna applicator of the first channel is applied to the output of the receiving channel switch for 80% of its duration. The signal from the output of the receiving channel switch is fed through the circulator to the receiver input for amplification and final processing. After the period receiving the signal from the antenna-applicator of the first channel expires, for the remaining part of 20% of the duration of the sub-period, the control code of the switching signal generator is applied to the control input of the switching signal switch to the input of the circulator, after passing in the direction of circulation, it enters the input of the receiving channel switch. The receiving channel switch has been open since thus period, the input of the first receiving channel is connected to the output of the receiving channel switch, and the noise generator signal is not reflected from the switch input, but passes through it and reaches the point of contact of the applicator antenna with the human body. If the antenna-applicator is perfectly aligned with the human body, then the entire electromagnetic wave of the noise generator will be radiated by the antenna-applicator and absorbed into the body. In this case, there will be no reflection from the antenna-applicator–body section and the same power will be recorded at the receiver output as in the first part of the second sub-period. If the antenna applicator is not aligned with the body, the electromagnetic wave generator noise impact from the section of the antenna applicator–the body and the reflected wave with the wave from the human body through the switch receiving channels and the circulator will be passed to the input of a low noise amplifier. Only part of the electromagnetic wave of the noise generator will be reflected from the antenna-applicator–body boundary if antenna is partly matching. It is possible to calculate the reflection coefficient from the antenna-applicator–body boundary by measuring the magnitude of the reflected wave. According to the fundamental principle of reciprocity, the reflection coefficients from the side of the human body and from the side of the antenna-applicator are equal. Therefore, defining the reflection coefficient body antenna applicator can be taken into account which part of the electromagnetic wave radiated by the body is reflected off of the body section–antenna applicator, which according to the experimental data permits us to determine the brightness temperature of the body tissues with one tenth of a degree accuracy when the reflection coefficient does not exceed 30%. For all other receiving channels, from the second to the fifth, the modulation control is similar. Also, the control and signal processing are similar for both frequency bands. The waveform of the analog signal from the output of the power detector is shown in Figure 4. The photo of a multichannel two-frequency radiothermograph located on the experimental stand is shown in Figure 5. The analog signal after the power detector stage is converted to digital form and processed according to the well-known algorithm [12] to calculate the radio brightness body temperatures for each channel and for each frequency range.

## 3. Results

A multi-channel multi-frequency radiothermograph receives signals from antennas-applicators located on the surface of the human body.

It is very important to choose the antenna-applicator location correctly in the installation method on the human body. Usually, the antennas are installed using a special holder with the guiding rods. Different holders are used depending on the organ being tested. A holder resembling a crown is used for the head internal temperature measurements (see Figure 5), which is fixed to the back of the chair the patient is sitting on. Shifting the sensor along the rod permits control of the coordinate position of the sensor, and moving the entire rod in a perpendicular direction permits to control another coordinate position. The sensor moves in height until it will comes into contact with the body. The sensor pressing force is determined by the degree of compression of the spiral spring installed at the end of the cylinder inside which the sensor rod can move. The antenna position in three coordinates is fixed by locking bolts. The sensor ensures reliable contact with the body due to spiral springs, even when the patient slightly changes the body position during the measurements.

The sensor’s choice of location depends on the organ being examined and the type of analysis being performed. Sometimes the analysis is carried out periodically for several days to follow the dynamics of the pathological process changes. In this case it is very important to ensure that the antennas are installed in the same positions every time. This can be done by using a template on a transparent film with marks of the sensor installation locations.

Each sensor has an individual number. The doctor chooses the organ and the analysis type that the patient needs from the program menu before measurements start. In response, the doctor receives a diagram indicating the sensor numbers and their installation locations. The program allows to enter the sensors new locations data and save this data for future use in case when sensors rearrange is necessary. Only correct sensors installation synchronized with the program data can guarantee the correct data interpolation process and their 3D visualization. Since there are only five sensors for one frequency channel, usually four sensors are located at the corners of the conditional square and one sensor in the center of the square. The sensors of the second frequency channel are located as close as possible to the sensors with the same numbers of the first frequency channel.

The sensors are installed so that the central sensor is located exactly at the site of the pathology, in case the pathology location is known in advance. It is more convenient to estimate the size of the pathological area and it depth using this arrangement. Several tests are performed with different sensor locations until one of the sensors will detects the location of the pathology in case the location of the pathology is not known in advance.

After processing the signals from one antenna, the temperature profile of the human body is restored along the normal line at the antenna applicator installation point. To restore the three-dimensional distribution function of the internal body temperatures the mathematical interpolation data procedure is used. It should be noted that the data obtained, as a rule, represent the body’s thermal response to various physiological effects—glucose test, laser or microwave local heating, exposure to drugs, and others. The results of the full-scale analysis are a three-dimensional function of internal temperatures that changes over time throughout the analysis. After the analysis, the obtained dynamic three-dimensional temperature distribution function is analyzed by the doctor to make or clarify the diagnosis, localization of the site of the inflammatory process, malignant tumor or thermal anomaly. Previously, when using a single-frequency radiothermograph, the measurement results were presented as a two-dimensional image on a monitor screen or a printout with the temperature values displayed in pseudo-colors. Images could be viewed on the screen of a personal computer monitor as a movie that could be stopped or accelerated at any time. Such a method of data visualization for multi-channel multi-frequency radiothermography is not applicable. Therefore, it is necessary to develop a new method of data visualization that allows analyzing three-dimensional images of the thermal field of the human body for more accurate localization of pathologies with the determination both their location and the depth.

Various methods of visualization of 3D objects are known, for example, holographic [44], interferometric [45], stereoscopic [46], and others. All these methods require special equipment and display only a 3D image of the external surface of the object, although with the illusion of volume. For opaque objects, displaying the internal structure of the object is not intended to use such methods.

To represent the distribution of a three-dimensional temperature function in a three-dimensional space on a flat screen the special algorithms and programs are needed. The classical methods of modeling 3D graphics used in 3D graphics editors (3D Max, ZBrush, Blender, etc.) are not suitable in this case, because they are based on modeling the boundaries of 3D objects, and in this case it is necessary to observe the internal structure of the object, which is hidden behind the boundaries. In addition, the temperature distribution boundary is not determined. Therefore, it is advisable to use slices to determine the internal structure of the temperature distribution. If, for example, during tomography, the slice pattern is determined by the device, in this case, the temperature value on the slice will be determined by double interpolation-linear and two-dimensional means.

A 3D structure for determining the temperature values on the slice is shown in the Figure 6.

Points A, B, C, D, E are located on the surface of the patient’s body. They contain the radiothermograph applicator antennas that allow to determine the surface temperature at these points and the temperature at a certain depth at points A’, B’, C’, D’, E’. The slice is defined by the abcd plane and is located at the depth of interest. The temperature at points a, b, c, d, and e is determined by linear interpolation between points AA’, BB’, CC’, DD’, and EE’, respectively.

The analysis of various methods and numerical experiments show that the interpolation method is the most suitable to use, which is based on a mathematical model of an elastic thin plate bent under the influence of external forces applied at points (*x_i_*, *y_i_*), *i* = 1, *N*. The Equations (3) and (4) of the total free energy of a curved elastic plate gives a relation describing a two-dimensional spline surface:(3)$$\varphi (x,y)=\sum_{i=1}^{N}C_{i}[{\left(x_{i}-x\right)}^{2}+{\left(y_{i}-y\right)}^{2}]\times \ln[{\left(x_{i}-x\right)}^{2}+{\left(y_{i}-y\right)}^{2}]+Ax+By+D;$$

The coefficients are determined from the formulas:(4)$$\begin{array}{l} \varphi (x_{i},y_{i})=T_{i},\;i=(1,N);\\ \sum_{i=1}^{N}C_{i}=0;\\ \sum_{i=1}^{N}C_{i}x_{i}=0;\\ \sum_{i=1}^{N}C_{i}y_{i}=0; \end{array}$$

(*x_i_*,*y_i_*)—points a,b,c,d,e. *T_i_*—temperature at these points.

We obtain *N* + 3 equations with *N* + 3 unknowns. In our case, *N* = 5.

For clarity, here is an example of interpolation of a simple Formula (5):(5)$$z(x,y)={\left(x-0.5\right)}^{2}+{\left(y-0.5\right)}^{2}$$
two methods for 10 random points.

The map of the original function and 10 random points that are used for interpolation is shown in Figure 7a. The result of the inverse distance interpolation, in which the weight of each of the 10 points is inversely proportional to the square of the distance to the interpolated point, is shown in Figure 7b. The result of the interpolation by the proposed method is shown in Figure 7c. It can be seen that the map obtained by this method most closely looks like the map of the original function.

The special program based on the interpolation algorithm described above was developed to display the temperature of a certain area inside the human skull. The layout of the sensors is similar to their location in Figure 6. The human head model was created using a 3D graphics editor. On the monitor screen, the doctor can see one of the two images shown in Figure 8. The schematic representation of a human head with five sensors installed and three surfaces with color temperature markings on them is shown in Figure 8a. The color palette for the temperature is on the left. The upper and lower surfaces are stationary, and the middle surface can be moved either with the keys or with the mouse wheel. In this case, the temperature on this surface is automatically recalculated and displayed as a pseudo-colors. The image can be rotated arbitrarily around the *x*, *y*, and *z* axes for ease of perception. The image of sensors and any surfaces can be disabled. The temperature distribution on the middle plane, as well as the depth of the plane, is shown in Figure 8b. In this form, all the data obtained during the analysis are available. Presented in this way, the dynamic data of the 3D distribution of the internal temperatures of the human body allow the doctor to more accurately determine the location of the pathology in its presence.

The experimental stand with the five-channel two-band radiothermograph was assembled for verification of the results obtained and debugging the algorithms and software for 3D visualization of the internal thermal field of the human body. A plastic bag with saline solution was used as a body simulator for testing and calibration (Figure 5). Test measurements were carried out directly on the human body after radiothermograph calibration. The results of measuring the thermal field of the human head obtained using a five-channel dual-band radiothermograph at frequencies F1 and F2 are shown in Figure 9a,b.

Similar results were obtained for the chest in Figure 10a,b. The results obtained will be used to calculate and visualize a 3D model of the thermal field in the chest and skull area similar to those shown in Figure 8. The internal temperature displaying the intermediate plane, whose position is specified by using the wheel of a computer mouse wheel, will more accurately determine the position of a malignant tumor, given the depth of the location. In addition, you can quickly view the results of a long test in the “cartoon” mode to analyze the dynamic thermal processes inside the human body.

## 4. Discussion

Body temperature is the most important indicator of human health. Usually the surface temperature of the body is measured using contact thermometers or non-contact infrared pyrometers for general diagnostic purposes. It is possible to significantly increase the list of various detectable diseases then measure the subsurface internal body temperature. Measuring the internal body temperature using ordinary invasive sensors gives a significant error, since the introduction of the sensor under the skin causes a violation of the internal heat field. It is possible to measure the internal temperature of the body non-invasively, by measuring the power of radio thermal radiation emitted by the internal body tissues and reaching its surface in the microwave range of electromagnetic waves. Microwave radiometers with ultra-sensitive receivers are used to measure the power of radio thermal radiation in the microwave range.

The use of microwave radiometers in medicine required the development of special antennas applicators and the improvement of methods for receiving and processing radio waves. The improvement of radiometer technology has made it possible to design multi-channel radiothermographs capable of measuring the internal temperature of the body simultaneously at several points of the body. Also were developed multi-frequency radiothermographs capable of receiving electromagnetic waves emitted by the tissues of the human body at different depths. Electromagnetic waves of different lengths have different attenuation values characteristics inside the body due to the well-known skin effect. Measuring the brightness temperatures of the body at different wavelengths permits the temperature profile to be calculated in depth.

This article presents the results of the development of a new radiothermograph that combines both the advantages of multi-channel and multi-frequency. Such a radiothermograph gives an opportunity to calculate and 3D visualize the internal thermal field of a human body. The well-known method of data interpolation, previously used in remote-sensing systems, as well as the method of visualizing the internal thermal field using a moving plane, were used in the new radiothermograph.

The new radiothermograph has been tested on the test objects and on the human body. The detailed information about the internal thermal field of a human body has made possible not only detection of an internal malignant tumor, but also determination of the depth of its location. The new radiothermograph gives an opportunity for more detailed analysis of the state of the vascular system of the brain and early diagnosis of various brain pathologies. Of course, it is too early to talk about all the possibilities and advantages of the new radiothermograph, which will be confirmed during clinical tests that will begin in the nearest future.

The new radiothermograph for 3D visualization of the internal thermal human body field has a minimal configuration and is intended rather to demonstrate the capabilities of the method. For effective use in clinical practice, five receiving channels and two frequency bands are absolutely not enough. Thus, when analyzing for the detection of the most common malignant pathology—women breast cancer, a complete analysis of one breast for sufficient reliability, a five-channel radiothermograph requires at least two analyses. For a complete analysis on two mammary glands simultaneously, the radiothermograph must have at least 20 receiving channels in at least three frequency bands.

For further research, it is necessary to solve the problems of developing broadband antennas for several frequency bands and micro miniature receivers combined with antennas. As well as improving the methods of clinical application and collecting a database for further use of artificial intelligence in order to analyze data and make recommendations to the doctor based on previously accumulated experience.

## 5. Conclusions

The relationship between the maximum detection depth of a local heat source inside the human body with the magnitude of the skin-layer, the sensitivity of the radiometer and the value of the thermal contrast of the source compared to other tissues were obtained as a result of the development and simulation. A number of experiments were carried out on the equivalent of the human body and directly on the real human body in order to obtain data for further 3D interpolation and visualization of the structure of the internal thermal field inside the human body on a computer monitor screen using the developed five-channel dual-frequency radiothermograph. The results obtained will be used for further improvement of the methods of multichannel multi-frequency radiothermograph applications in medical practice for more accurate localization of pathological neoplasms inside the human body.

## 6. Patents

Author’s certificate No. 2020667485 for “Program control, pre-processing and data transmission for multi-channel radiothermograph” and author’s certificate No. 2020667484 for “Program the radiothermograph data processing and visualization for a human head” were issued for software that has been designed during this work.

## Figures and Tables

**Figure 1 sensors-21-04005-f001:**
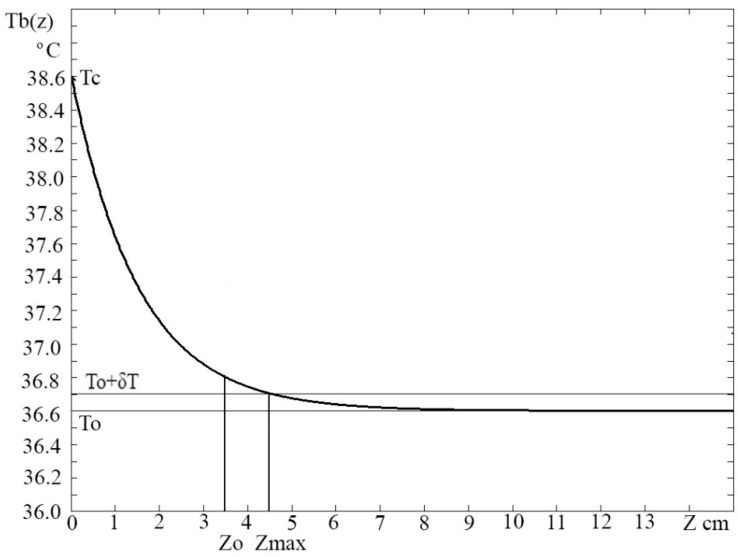
Calculated values of brightness temperatures at the location of the tumor, depending on the depth of the tumor in cm for the skin layer of 3.5 cm with a thermal contrast of the tumor of 2 degrees.

**Figure 2 sensors-21-04005-f002:**
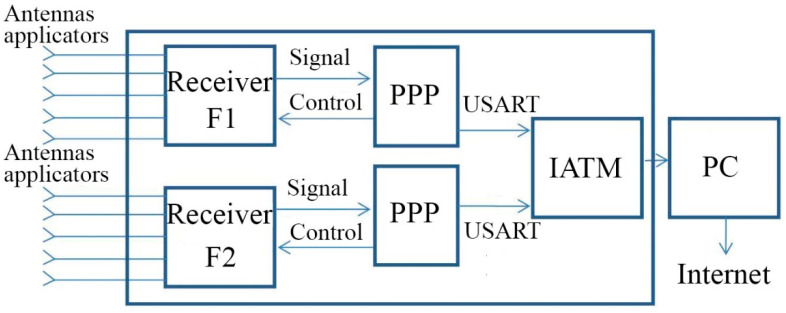
Block diagram of a multichannel radiothermograph.

**Figure 3 sensors-21-04005-f003:**
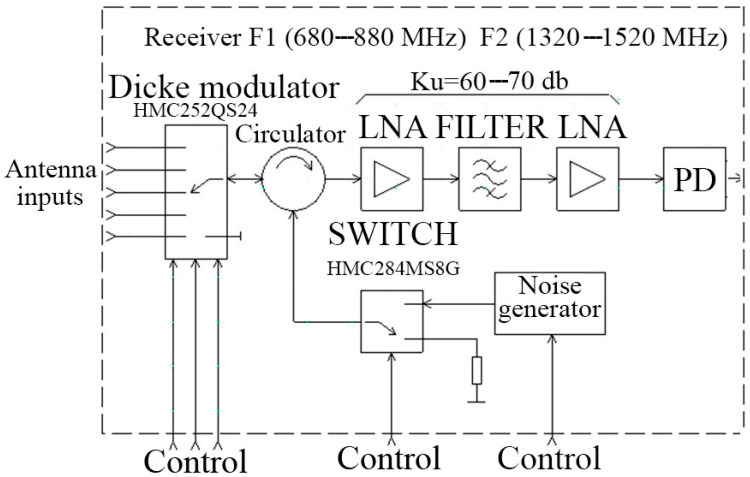
Block diagram of a five-channel radiometer.

**Figure 4 sensors-21-04005-f004:**
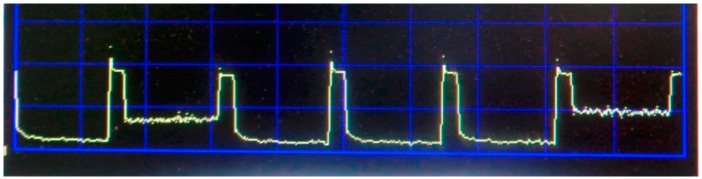
Waveform of the signal at the output of the power detector.

**Figure 5 sensors-21-04005-f005:**
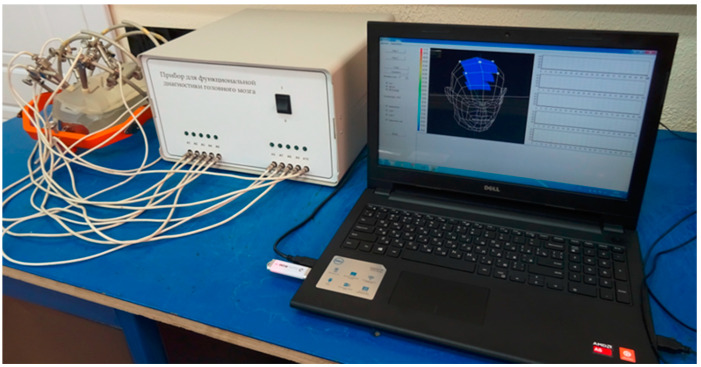
Appearance of a multichannel radiothermograph.

**Figure 6 sensors-21-04005-f006:**
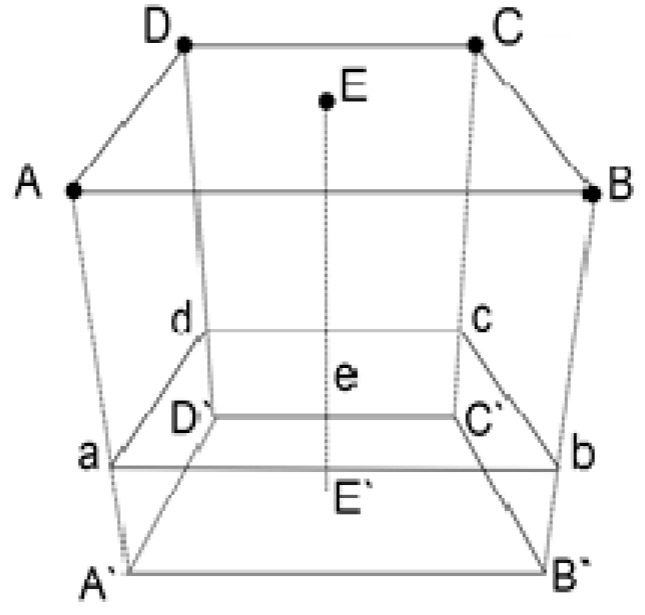
Three-dimensional (3D) structure for determining the temperature values on the slice.

**Figure 7 sensors-21-04005-f007:**
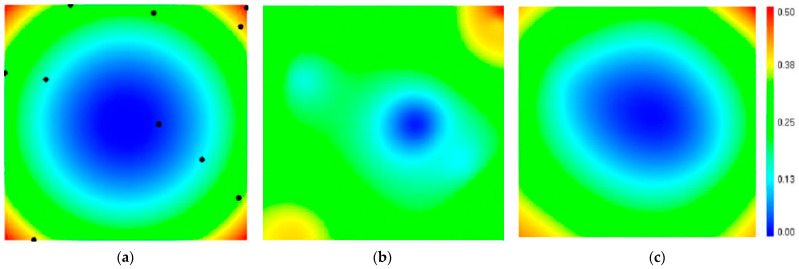
Example of function interpolation using two methods: (**a**) shows a map of the original function; (**b**) shows the result of inverse distance interpolation; (**c**) shows the result of the interpolation.

**Figure 8 sensors-21-04005-f008:**
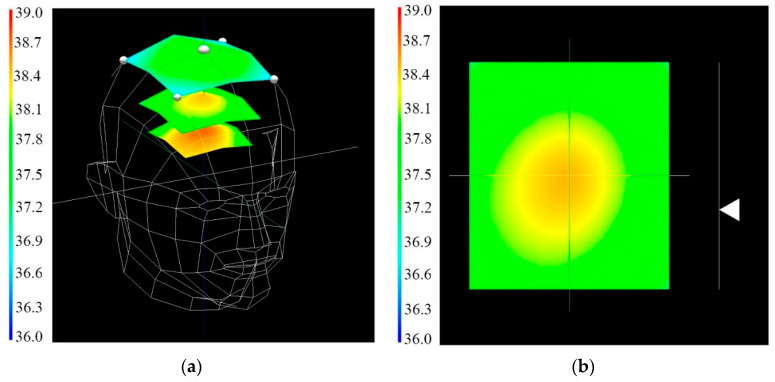
(**a**) The radiothermography results on the screen. (**b**) The temperature distribution on the middle plane.

**Figure 9 sensors-21-04005-f009:**
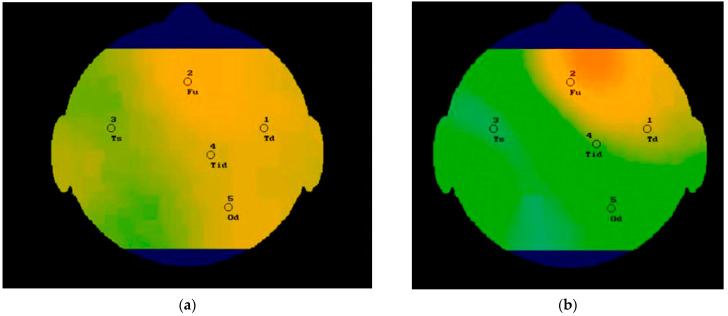
The result of an experimental research of the human head. At the frequency F1—(**a**) and F2—(**b**).

**Figure 10 sensors-21-04005-f010:**
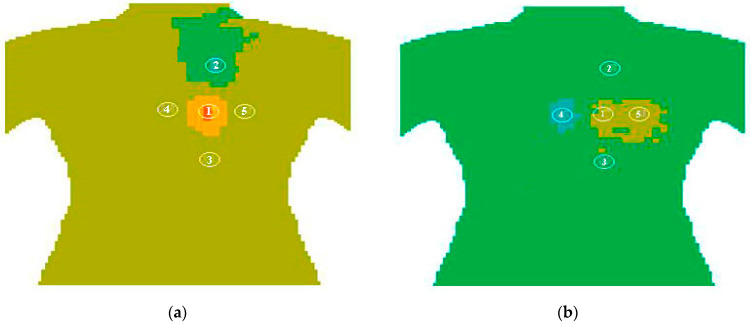
The result of experimental research of the human chest. At the frequency F1—(**a**) and F2—(**b**).

## Data Availability

All data is contained within the article no data from external sources were used.

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
