# Peer review of "Measurement and 3D Visualization of the Human Internal Heat Field by Means of Microwave Radiometry"

_sensors, 2021, doi:10.3390/s21124005_

Round 1

Reviewer 1 Report

The article is too long. Lines 251 to 256 talk about the linear interpolation of the temperature distribution between the two extreme surfaces. The temperature change along the depth of penetration into the tissue is nonlinear. The formulas and drawings in points 3 of the article are poorly described.

Author Response

  1. The article is too long.

The article describes results of the work for two years. It is review including hard, soft and tests results. If we would delete something the article would be incomplete.

  1. Lines 251 to 256 talk about the linear interpolation of the temperature distribution between the two extreme surfaces. The temperature change along the depth of penetration into the tissue is nonlinear.

Yes, we use linear interpolation. Of course, real temperature distribution is non-linear.

But we have two measured by radiometer temperatures at two frequencies only (and one temperature from skin sensor).

Linear interpolation is most suitable for interpolation between two points. If we would have three or five frequencies we would use more accurate cube or five power interpolation polynomial function.

  1. The formulas and drawings in points 3 of the article are poorly described

Yes. But we used well known interpolation method described in scientific literature.

(Nesterenko EA. The ability of a spline surface using to build surfaces based on the results of surveys. Notes of the Mining Institute, 2013, V.204, pp. 127-133)

Unfortunately, this publication has down in Russian and not accessible in the Internet so we did not include it in References.

Reviewer 2 Report

The authors have presented an excellent paper. However, some minor issues must be better explained before publication.

  • The mathematical formalism must be improved. For instance, eq (2), what is \alpha? How is the skin layer size defined? Where are F1 and F2 used in the equation?
  • Figure 2 must be improved. The purpose of such a figure is to demonstrate a logical sequence; it is not necessary to include a power supply. Moreover, it should be interesting to explain how the sensors are distributed around the area of interest. Is there a difference in the results, or is the system insensitive to it?
  • The authors must carefully think about their focus on the paper. Is the focus the measurement system, the software methodology, or the results obtained by it? For instance, figure (3) shows the block diagram of the proposed radiometer. If this is the focus, it should be better described, compared, etc... It is the same for the other approaches. The authors have presented a high-level view without entering in details of any specific part. I understand that it is a multi-dimensional solution. However, the main part must be fully explained. There are so many open questions that it is even challenging to report. 
  • The visualization part should be removed as a contribution (it should be presented as a tool). There are several ways of doing that and, to proper academic research the authors should have commented, discussed, and justified their approach. Moreover, this part has no novelty, so it is not worth the effort.
  • The discussion follows the former comment about focus. Many good things are going on in this paper, but none were adequately discussed in a formal academic way.

All in all, it is an excellent paper, with the potential to be broken into at least 2 works (hardware and software). Put all in one publication leads to a situation where the main topics are presented at a high level of demonstration more than a research. 

Author Response

  1. The mathematical formalism must be improved. For instance, eq (2), what is \alpha? How is the skin layer size defined? Where are F1 and F2 used in the equation?

The mathematical formalism described in details in point [3] of Referenses (E.P. Novichikhin, I.A. Sidorov, V.Yu. Leushin, S.V. Agasieva, S.V. Chizhikov. Detection of a local source of heat in the depths of the human body by volumetric radiothermography. RENSIT, 2020, Vol 12(2), pp. 305-312. http://rensit.ru/vypuski/article/332/12(2)1-312.pdf DOI: 10.17725/rensit.2020.12.305.)

In current article we represent the resulting formulas only.

The “alpha” parameter was introduced to simplify the formula (2) representation.

Formula (2) without “alpha” parameter will look like: (in manuscript file)

The skin-effect is well known. The skin layer size formula can be find for instance on reference: https://en.wikipedia.org/wiki/Skin_effect

F1 and F2 – are the middle frequencies for receiver’s band. The frequency of electro-magnetic wave F and its wave length λ defined by formula λ=c/F, were c is lite velocity. In all formulas above the parameters with index λ1 correspond to F1 frequency and with index λ2 correspond to F2 frequency. So F1 and F2 presents in formulas indirectly.

  1. Figure 2 must be improved. The purpose of such a figure is to demonstrate a logical sequence; it is not necessary to include a power supply. Moreover, it should be interesting to explain how the sensors are distributed around the area of interest. Is there a difference in the results, or is the system insensitive to it?

Figure 2 has been improved.

Power supply unit was removed from picture.

The sensors arrangement is interesting and important question. We made special research to clarify this question. The results of this research will be the topic of one of future articles. Briefly the sensors arrangement depend on many factors like organ for analysis, the hairs presence the contact with human skin and so on…

  1. The authors must carefully think about their focus on the paper. Is the focus the measurement system, the software methodology, or the results obtained by it? For instance, figure (3) shows the block diagram of the proposed radiometer. If this is the focus, it should be better described, compared, etc... It is the same for the other approaches. The authors have presented a high-level view without entering in details of any specific part. I understand that it is a multi-dimensional solution. However, the main part must be fully explained. There are so many open questions that it is even challenging to report.

Yes. All aspects of multi channels and multi frequencies radiothermography impassible to describe in single article. The main purpose of this article is to represent the main results of two years research devoted to design hard and soft and tests results of new multi channels and multi frequencies radio thermograph. In article [1] of References described method of “sensing depth of Microwave Radiation for internal body temperature measurement” using three different frequencies bands. In article [20] of References described method of sensing human body using many channels working on the same frequency. The novelty of current article that we combined this two approaches in our new radiothermograph.

Of course, there are so many important questions…

For instance:

  • The calibration questions. Internal and external calibration of each sensor on temperatures and depth;
  • Coherence questions between antennas and human body which cause partly reflection of wave from antenna. To regard this reflection we use special procedure not described in current article;
  • The external distortions influence and methods of their reduction.

All this questions can be the topics of future articles.

  1. The visualization part should be removed as a contribution (it should be presented as a tool). There are several ways of doing that and, to proper academic research the authors should have commented, discussed, and justified their approach. Moreover, this part has no novelty , so it is not worth the effort.

The visualization method used has no novelty, but application this method to display the internal human body thermal field using multi channels and multi frequencies radiothermograph data is new.

  1. The discussion follows the former comment about focus. Many good things are going on in this paper, but none were adequately discussed in a formal academic way.

Our research is not finished yet. Current article should be regarded as “communications” article but not a “research article”.

Reviewer 3 Report

A Manuscript titled "Measurement and 3-D Visualization of the Human Internal Heat Field by Means of Microwave Radiometry" has been presented. Following suggestions will be helpful to further improver the manuscript.

  • In-text citation numbering format should be consistent (i.e [1-17], use [38-40] for [38,39,40], use [12,13,20-22] for [12,13,20,21,22], and similarly check for others)
  • Add reference for equations (where possible)
  • Figure 1, axis and text is not readable, need to be improved.
  • Adjust the size for text in Fig. 2 similar to text size in Fig. 3 for better readability.
  • Figure 8, increase the scale text size, its not readable.
  • English check for typos and grammatical errors recommended.

Author Response

  1. In-text citation numbering format should be consistent (i.e [1-17], use [38-40] for [38,39,40],use [12,13,20-22] for [12,13,20,21,22], and similarly check for others)

Has done

  1. Add reference for equations (where possible)

Has done

  1. Figure 1, axis and text is not readable, need to be improved.

Has done

  1. Adjust the size for text in Fig. 2 similar to text size in Fig. 3 for better readability.

Has done

  1. Figure 8, increase the scale text size, its not readable.

Has done

Round 2

Reviewer 2 Report

from my opnion, the paper can now be published.

Author Response

We would like to express our gratitude to the anonymous reviewer whose comment helped us to improve the overall quality of our manuscript.